# Dynamic Learning Rate for Deep Reinforcement Learning: A Bandit Approach

## Abstract

In Deep Reinforcement Learning models trained using gradient-based techniques, the choice of optimizer and its learning rate are crucial to achieving good performance: higher learning rates can prevent the model from learning effectively, while lower ones might slow convergence. Additionally, due to the non-stationarity of the objective function, the best-performing learning rate can change over the training steps. To adapt the learning rate, a standard technique consists of using decay schedulers. However, these schedulers assume that the model is progressively approaching convergence, which may not always be true, leading to delayed or premature adjustments. In this work, we propose dynamic Learning Rate for deep Reinforcement Learning (LRRL), a meta-learning approach that selects the learning rate based on the agent's performance during training. LRRL is based on a multi-armed bandit algorithm, where each arm represents a different learning rate, and the bandit feedback is provided by the cumulative returns of the RL policy to update the arms' probability distribution. Our empirical results demonstrate that LRRL can substantially improve the performance of deep RL algorithms.

## 1 Introduction

Reinforcement Learning (RL), when combined with function approximators such as Artificial Neural Networks (ANNs), has shown success in learning policies that outperform humans in complex games by leveraging extensive datasets (see, *e.g.*, Silver et al., 2016; Lample & Chaplot, 2017; Vinyals et al., 2019; Wurman et al., 2022). While ANNs were previously used as value function approximators (Riedmiller, 2005), the introduction of Deep Q-Networks (DQN) by Mnih et al. (2013; 2015) marked a significant breakthrough by improving learning stability through two mechanisms: the target network and experience replay.

The experience replay (see Lin, 1992) stores the agent's interactions within the environment, allowing sampling of past interactions in a random way that disrupts their correlation. The target network further stabilizes the learning process by periodically copying the parameters of the learning network. This strategy is crucial because the Bellman update —using estimations to update other estimations— would otherwise occur using the same network, potentially causing divergence. By leveraging the target network, gradient steps are directed towards a periodically fixed target, ensuring more stability in the learning process. Additionally, the learning rate hyperparameter controls the magnitude of these gradient steps in optimizers such as the stochastic gradient descent algorithm, affecting the training convergence.

The learning rate is one of the most important hyperparameters, with previous work demonstrating that decreasing its value during policy finetuning can enhance performance by up to 25% in vanilla DQN (Agarwal et al., 2022). Determining the appropriate learning rate[1] is essential for achieving good model performance: higher values can prevent the agent from learning, while lower values can lead to slow convergence (see Goodfellow et al., 2016; Blier et al., 2019; You et al., 2019). However, finding a learning rate value that improves the model performance requires extensive and computationally expensive testing. In order to adapt its initial choice during training, optimizers such as

---

[1]The terms "learning rate" and "step-size" are often used interchangeably in the literature and they technically refer to the same concept.

Adam (Kingma & Ba, 2015) and RMSProp (Tieleman & Hinton, 2012) employ an internal scheme that dynamically adjusts the learning rate, considering, for instance, past gradient information. Nevertheless, various learning rate scheduling strategies can be combined with the optimizer to decrease the learning rate and improve the convergence over the training steps.

Standard learning rate schedulers typically decrease the learning rate based on training progress using, *e.g.*, linear or exponential decay strategies (Senior et al., 2013; You et al., 2019). In the context of RL, this approach can lead to premature or delayed learning rate adjustments, which may hinder the agent's ability to learn. Unlike supervised learning, RL usually involves generating data by trading off between exploration (discovery of new states) and exploitation (refining of the agent's knowledge). As the policy improves, the data distribution encountered by the agent becomes more concentrated, but this evolution occurs at a different pace than the overall training progress. For instance, some environments require extensive exploration due to the sparseness of rewards, while others need more exploitation to refine the policy to the complexity of the task. Consequently, a more sophisticated decaying learning rate strategy that accounts for policy performance rather than training steps can significantly enhance learning in deep RL.

In this work, we propose dynamic **L**earning **R**ate for deep **R**einforcement **L**earning (LRRL), a method to select the learning rate *on the fly* for deep RL. Our approach acknowledges that *different learning phases require different learning rates*, and as such, instead of scheduling the learning rate decay using some blanket approach, we dynamically choose the learning rate using a Multi-Armed Bandit (MAB) algorithm, which accounts for the current policy's performance. Our method has the advantage of being algorithm-agnostic and applicable to any optimizer, although the results show that it works best when coupled with Adam. We conduct experiments on our approach using baselines provided in the Dopamine framework (see Castro et al., 2018). Our results focus on exploiting different settings for LRRL to illustrate its robustness under many possible configurations. Our main contributions are the following:

- We introduce LRRL, the first approach, to our knowledge, that leverages a multi-armed bandit algorithm to select the learning rate dynamically in deep RL. Our results demonstrate that LRRL achieves competitive performance with or superior to standard deep RL algorithms using fixed baselines or traditional learning rate schedulers.

- Our results show that LRRL significantly reduces the need for hyperparameter optimization by dynamically selecting from a set of possible learning rates using a multi-armed bandit approach. This method mitigates the need for exhaustive techniques like grid search, as it efficiently adapts the learning rate during training in a single run, rather than requiring multiple runs to test each learning rate individually.

- We assess the robustness of our method by employing the Adam and RMSProp optimizers with different sets of arms. We also compare the results using stochastic and adversarial multi-armed bandit algorithms in Appendix A.2.

## 2  RELATED WORK

**Multi-armed bandit for (hyper)parameter selection.** Deep RL is known to be overly optimistic in the face of uncertainty (Ostrovski et al., 2021), and many works have addressed this issue by proposing conservative policy updates (van Hasselt et al., 2016; Fujimoto et al., 2019; Agarwal et al., 2020). However, when the agent is able to interact with the environment, this optimism can encourage exploration, potentially leading to the discovery of higher returns. Building on this idea, Moskovitz et al. (2021) use an adversarial MAB algorithm to trade-off between pessimistic and optimistic policy updates based on the agent's performance over the learning process. In order to select the learning rate for stochastic gradient MCMC, Coullon et al. (2023) employ an algorithm based on Successive Halving (Karnin et al., 2013; Jamieson & Talwalkar, 2016), a MAB strategy that promotes promising arms and prunes suboptimal ones over time. In the context of hyperparameter optimization, Successive Halving has also been used in combination with infinite-arm bandits to select hyperparameters for supervised learning (Li et al., 2018; Shang et al., 2019). A key difference between these approaches to hyperparameter optimization for supervised learning and our work is that we focus on selecting the best learning rate from a predefined set, rather than performing an extensive and computationally expensive search over the hyperparameter space. Close to our work,

Liu et al. (2020) propose Adam with Bandit Sampling (ADAMBS), which employs the Exponential-weight algorithm for Exploration and Exploitation (Auer et al., 2002) to enhance sample efficiency by incorporating importance sampling within the Adam optimizer. While ADAMBS prioritizes informative samples, our method focuses on RL tasks by dynamically adjusting the learning rate, accelerating learning when the policy is far from optimal, and slowing it down as it converges.

**Learning rate adapters/schedulers.** Optimizers such as RMSProp (Tieleman & Hinton, 2012) have an adaptive mechanism to update a set of parameters $\theta$ by normalizing past gradients, while Adam (Kingma & Ba, 2015) also incorporates momentum to smooth gradient steps. However, despite their widespread adoption, these algorithms have inherent limitations in non-stationary environments since they do not adapt to changes in the objective function over time (see Degris et al., 2024). Increment-Delta-Bar-Delta (IDBD), introduced by Sutton (1992), has an adaptive mechanism based on the loss to adjust the learning rate $\eta_i$ for each sample $x_i$ for linear regression and has been extended to settings including RL (Young et al., 2019). Learning rate schedulers with time decay (Senior et al., 2013; You et al., 2019) are coupled with optimizers, assuming gradual convergence to a good solution, but often require task-specific manual tuning. A meta-gradient reinforcement learning is proposed in Xu et al. (2020), composed of a two-level optimization process: one that uses the agent's objective and the other to learn meta-parameters of the objective function. Our work differs from these methods by employing a multi-armed bandit approach to dynamically select the learning rate over the training process, specifically targeting RL settings.

## 3 PRELIMINARIES

This section introduces the Reinforcement Learning and Multi-Arm Bandits frameworks, defining supporting notation.

### 3.1 DEEP REINFORCEMENT LEARNING

An RL task is defined by a Markov Decision Process (MDP), that is by a tuple $(\mathcal{S}, \mathcal{A}, P, R, \gamma, T)$, where $\mathcal{S}$ denotes the state space, $\mathcal{A}$ the set of possible actions, $P : \mathcal{S} \times \mathcal{A} \times \mathcal{S} \to [0, 1]$ the transition probability, $R : \mathcal{S} \times \mathcal{A} \to \mathbb{R}$ the reward function, $\gamma \in [0, 1]$ the discount factor, and $T$ the horizon length in episodic settings (see, *e.g.*, Sutton & Barto, 2018 for details). In RL, starting from an initial state $s_0$, a learner called *agent* interacts with the environment by picking, at time $t$, an action $a_t$ depending on the current state $s_t$. In return, it receives a reward $r_t = R(s_t, a_t)$, reaching a new state $s_{t+1}$ according to the transition probability $P(s_t, a_t, \cdot)$. The agent's objective is to learn a policy $\pi : \mathcal{S} \times \mathcal{A} \to [0, 1]$ which maps a distribution of actions given the current state, aiming to maximize expected returns $Q^\pi(s, a) = \mathbb{E}_\pi\left[\sum_{t=0}^T \gamma^t r_t \mid s_0 = s, a_0 = a\right]$.

To learn how to perform a task, value function-based algorithms coupled with ANNs (Mnih et al., 2013; 2015) approximate the *quality* of a given state-action pair $Q(s, a)$ using parameters $\theta$ to derive a policy $\pi_\theta(s) = \arg\max_{a \in \mathcal{A}} Q_\theta(s, a)$. By storing transitions $(s, a, r, s') = (s_t, a_t, r_t, s_{t+1})$ into the replay memory $\mathcal{D}$, the objective is to minimize the loss function $\mathcal{J}(\theta)$ defined by:

$$\mathcal{J}(\theta) = \mathbb{E}_{(s,a,r,s') \sim \mathcal{D}}\left[r + \gamma \max_{a' \in \mathcal{A}} Q_{\theta^-}(s', a') - Q_\theta(s, a)\right]^2, \tag{1}$$

where $\theta^-$ are the target network parameters used to compute the target of the learning network $y = r + \gamma \max_{a' \in \mathcal{A}} Q_{\theta^-}(s', a')$. The parameters $\theta^-$ are periodically updated by copying the parameters $\theta$, leveraging stability during the learning process by fixing the target $y$. The minimization, and hence, the update of the parameters $\theta$, is done according to the optimizer's routine. A simple possibility is to use stochastic gradient descent using mini-batch approximations of the loss gradient:

$$\theta_{n+1} \leftarrow \theta_n - \eta \nabla_\theta \mathcal{J}(\theta_n), \quad \text{where} \quad \nabla_\theta \mathcal{J}(\theta_n) \approx \frac{1}{|\mathcal{B}|} \sum_{(s,a,r,s') \in \mathcal{B}} 2\big(Q_\theta(s, a) - y\big)\nabla_\theta Q_\theta(s, a) \tag{2}$$

with $\mathcal{B}$ being a mini-batch of transitions sampled from $\mathcal{D}$ and $\eta$ is a single scalar value called the *learning rate*. Unlike in supervised learning, where the loss function $\mathcal{J}(\theta)$ is typically stationary, RL

presents a fundamentally different challenge: the policy is continuously evolving, leading to shifting distributions of states, actions, and rewards over time. This continuous evolution introduces instability during the learning process, which deep RL mitigates by employing a large replay memory and calculating the target using a *frozen* network with parameters $\theta^-$. However, stability also depends on how the parameters $\theta$ change during each update. This work aims to control these changes by dynamically selecting the learning rate $\eta$ over the training steps.

## 3.2 MULTI-ARMED BANDIT

Multi-Armed Bandits (MAB) provide an elegant framework for making sequential decisions under uncertainty (see for instance Lattimore & Szepesvári, 2020). MAB can be viewed as a special case of RL with a single state, where at each round $n$, the agent selects an arm $k_n \in \{1, \ldots, K\}$ from a set of $K$ arms and receives a feedback (reward) $f_n(k_n) \in \mathbb{R}$. Like RL, MAB algorithms must balance the trade-off between exploring arms that have been tried less frequently and exploiting arms that have yielded higher rewards up to time $n$.

To account for the non-stationarity of the RL rewards, we will consider in this work the MAB setting of adversarial bandits Auer et al. (2002). In this setting, at each round $n$, the agent selects an arm $k_n$ according to some distribution $p_n$ while the environment (the *adversary*) arbitrarily (*e.g.*, without stationary constraints) determines the rewards $f_n(k)$ for all arms $k \in \mathcal{K}$. MAB algorithms are designed to minimize the *pseudo-regret* $G_N$ after $N$ rounds defined by:

$$G_N = \max_{k \in \{1, \ldots, K\}} \mathbb{E} \left[ \sum_{n=1}^{N} f_n(k) - \sum_{n=1}^{N} f_n(k_n) \right],$$

where the randomness of the expectation depends on the MAB algorithm and on the adversarial environment, $\sum_{n=1}^{N} f_n(k)$ represents the accumulated reward of the single best arm in hindsight, and $\sum_{n=1}^{N} f_n(k_n)$ is the accumulated reward obtained by the algorithm. A significant component in the adversarial setting is to ensure that each arm $k$ has a non-zero probability $p_n(k) > 0$ of being selected at each round $n$: this guarantees exploration, which is essential for the algorithm's robustness to environment changes.

## 4 DYNAMIC LEARNING RATE FOR DEEP RL

In this section, we tackle the challenge of selecting the learning rate over the training steps by introducing a dynamic Learning Rate for deep Reinforcement Learning (LRRL). LRRL is a meta-learning approach designed to dynamically select the learning rate in response to the agent's performance. LRRL couples with stochastic gradient descent optimizers and adapts the learning rate based on the reward achieved by the policy $\pi_\theta$ using an adversarial MAB algorithm. As the agent interacts with the environment, the average of observed rewards is used as bandit feedback to guide the selection of the most appropriate learning rate throughout the training process.

## 4.1 SELECTING THE LEARNING RATE DYNAMICALLY

Our problem can be framed as selecting a learning rate $\eta$ for policy updates —specifically, when updating the parameters $\theta$ after $\lambda$ interactions with the environment.

Before training, a set $\mathcal{K} = \{\eta_1, \ldots, \eta_K\}$ of $K$ learning rates are defined by the user. Then, during training, a MAB algorithm selects, at every round $n$ —that is, at every $\kappa$ interactions with the environment— an arm $k_n \in \{1, \ldots, K\}$ according to a probability distribution $p_n$ defined based on previous rewards, as explained in the next section. The parameters $\theta$ are then updated using the sampled learning rate $\eta_{k_n}$. The steps involved in this meta-learning approach are summarized in Algorithm 1.

Note that the same algorithm might be used with learning rates schedulers, that is with $\mathcal{K} = \{\eta_1, \ldots, \eta_K\}$ where $\eta_k : \mathbb{N} \to \mathbb{R}_+$ is a predefined function, usually converging towards 0 at infinity. If so, the learning rate used at round $n$ of the optimization is $\eta_{k_n}(n)$.

---

**Algorithm 1** dynamic Learning Rate for deep Reinforcement Learning (LRRL)

---

**Parameters:**
  Set of learning rates $\mathcal{K} = \{\eta_1, \ldots, \eta_K\}$
  Number of episodes $M$
  Horizon length $T$
  Update window $\lambda$ for the learning network $\theta$
  Update window $\tau$ for the target network $\theta^-$
  Update window $\kappa$ for arm probabilities $p$
**Initialize:**
  Parameters $\theta$ and $\theta^-$
  Arm probabilities $p_0 \leftarrow (\frac{1}{K}, \ldots, \frac{1}{K})$
  MAB round $n \leftarrow 0$
  Cumulative reward $R \leftarrow 0$
  Environment interactions counter $C \leftarrow 0$
**for** episode $m = 1, 2, 3, \ldots, M$ **do**
    **for** timestep $t = 1, 2, 3, \ldots, T$ **do**
        Choose action $a_t$ following the policy $\pi_\theta(s)$ with probability $1 - \epsilon$ ▷ ($\epsilon$-greedy strategy)
        Play action $a_t$ and observe reward $r_t$
        Add $r_t$ to cumulative reward $R \leftarrow R + r_t$
        Increase environment interactions counter $C \leftarrow C + 1$
        **if** $C \mod \lambda \equiv 0$ **then**
            **if** $C \geq \kappa$ **then**
                Compute average of the last $C$ rewards $f_n \leftarrow \frac{R}{C}$
                Increase MAB round $n \leftarrow n + 1$
                Compute weights $w_n$ and arm probabilities $p_n$ using Equations (3, 4)
                Sample arm $k_n$ with distribution $p_n$
                Reset $R \leftarrow 0$ and $C \leftarrow 0$
            **end if**
            Update network parameter $\theta$ using the optimizer update rule with learning rate $\eta_{k_n}$
            Every $\tau$ steps update the target network $\theta^- \leftarrow \theta$
        **end if**
    **end for**
**end for**

---

### 4.2 Updating the Probability Distribution

As we expect that the agent's performance —and hence, the cumulative rewards— will improve over time, the MAB algorithm should receive non-stationary feedback. To take this non-stationary nature of the learning into account, we employ the Exponential-weight algorithm for Exploration and Exploitation (Exp3, see Auer et al., 2002 for an introduction). At round $n$, Exp3 chooses the next arm (and its associated learning rate) according to the arm probability distribution $p_n$ which is based on weights $(w_n(k))_{1 \leq k \leq K}$ updated recursively. Those weights incorporate a time-decay factor $\delta \in (0, 1]$ that increases the importance of recent feedback, allowing the algorithm to respond more quickly to improvements in policy performance.

Specifically, after picking arm $k_n$ at round $n$, the RL agent interacts $C$ times with the environment and the MAB algorithm receives a feedback $f_n$ corresponding to the average reward of those $C$ interactions. Based on Moskovitz et al. (2021), this feedback is then used to compute the *improvement in performance*, denoted by $f'_n$, obtained by subtracting the average of the past $j$ bandit feedbacks from the most recent one $f_n$:

$$f'_n = f_n - \frac{1}{j} \sum_{i=0}^{j-1} f_{n-i} \,.$$

The improvement in performance allows computation of the next weights $w_{n+1}$ as follows, where initially $w_1 = (0, \ldots, 0)$:

$$\forall k \in \{1, \ldots, K\}, \quad w_{n+1}(k) = \begin{cases} \delta \, w_n(k) + \alpha \, \frac{f'_n}{e^{w_n(k)}} & \text{if } k = k_n \\ \delta \, w_n(k) & \text{otherwise} \, , \end{cases} \quad (3)$$

where $\alpha > 0$ is a step-size parameter. The distribution $p_{n+1}$, used to draw the next arm $k_{n+1}$, is

$$\forall k \in \{1, \dots, K\}, \quad p_{n+1}(k) = \frac{e^{w_{n+1}(k)}}{\sum_{k'=1}^{K} e^{w_{n+1}(k')}} . \tag{4}$$

This update rule ensures that as the policy $\pi_\theta$ improves, the MAB algorithm continues to favor learning rates that are most beneficial under the current policy performance, thereby effectively handling the non-stationarity inherent in the learning process.

## 5 EXPERIMENTS

In the following sections, we investigate whether combining LRRL with Adam or RMSProp —two widely used optimizers— can improve cumulative returns in deep RL algorithms. To assess this, we compare LRRL against learning methods with and without schedulers using the baseline implementation of DQN provided in Dopamine (Castro et al., 2018). We test LRRL under different configurations and Atari games, reporting the average and standard deviation of returns over 5 runs. Details on the evaluation metrics and hyperparameters used in these experiments are summarized in Appendix B.

### 5.1 COMPARING LRRL WITH STANDARD LEARNING

In our first experiment, we consider a set of 5 learning rates, and compare the performance of 5 configurations of LRRL, each of them using a subset of those learning rates, against the DQN algorithm reaching best performance, in terms of maximum average return, among the 5 possible learning rates choices (see Figure 5 in Appendix A.1). More precisely, the set of learning rates is

$$\mathcal{K}(5) = \left\{ 1.5625 \times 10^{-5}, 3.125 \times 10^{-5}, 6.25 \times 10^{-5}, 1.25 \times 10^{-4}, 2.5 \times 10^{-4} \right\} .$$

The whole set $\mathcal{K}(5)$ is used by one LRRL version, while others are based on the 3 lowest ($\mathcal{K}_{\text{lowest}}(3)$), 3 middle ($\mathcal{K}_{\text{middle}}(3)$), 3 highest ($\mathcal{K}_{\text{highest}}(3)$) and 3 taking the lowest/middle/highest ($\mathcal{K}_{\text{sparse}}(3)$) learning rate values. All experiments use the Adam optimizer, and we report the return based on the same number of environment iterations.

Results are gathered in Table 1 and illustrated in Figure 1 along with four Atari games. They show that LRRL outperforms standard learning in two out of four tasks while remaining competitive in the others. Notably, LRRL not only matches or exceeds performance but also reduces the need for extensive parameter tuning by incorporating multiple learning rates in a single run. However, as the results indicate, the 3-arm bandit variants exhibit different behaviors during the learning process, suggesting that the choice of the number of arms and their values still requires task-specific tuning to achieve good performance.

| Game | DQN | LRRL $\mathcal{K}_{\text{lowest}}(3)$ | LRRL $\mathcal{K}_{\text{middle}}(3)$ | LRRL $\mathcal{K}_{\text{highest}}(3)$ | LRRL $\mathcal{K}_{\text{sparse}}(3)$ | LRRL $\mathcal{K}(5)$ |
|---|---|---|---|---|---|---|
| Asteroids | $1\,085 \pm 63$ | $1\,065 \pm 70$ | $1\,079 \pm 48$ | $1\,061 \pm 59$ | $1007 \pm 127$ | $1\,085 \pm 88$ |
| Breakout | $217 \pm 14$ | $220 \pm 24$ | $201 \pm 6$ | $177 \pm 8$ | $\mathbf{270} \pm 13$ | $253 \pm 20$ |
| Pong | $19 \pm 0.4$ | $18 \pm 0.3$ | $19 \pm 0.9$ | $19 \pm 0.5$ | $19 \pm 0.5$ | $19 \pm 0.8$ |
| Seaquest | $5\,881 \pm 1\,533$ | $5\,905 \pm 2\,557$ | $6\,135 \pm 2\,229$ | $6\,231 \pm 1\,802$ | $\mathbf{8\,920} \pm 2\,759$ | $6\,799 \pm 2\,060$ |

Table 1: Max average return (best in **bold** if significantly better than others) and its standard deviation for 4 Atari games.

To illustrate how LRRL adapts during training in response to non-stationary bandit feedback as policy performance improves, Figure 2 shows the systematic sampling of pulled arms (learning rates) and corresponding returns over training steps from a single run of LRRL $\mathcal{K}(5)$. In most of the tested environments, LRRL behaves similarly to time-decay schedulers by selecting higher learning rates during the early stages of training, gradually shifting toward arms with lower rates as training progresses. The exception is in the Pong environment, where the model converges after only a few iterations, resulting in a more uniform probability distribution across the set of arms.

### 5.2 COMBINING AND COMPARING SCHEDULERS WITH LRRL

Next, we consider using LRRL combined with learning rate schedulers. Specifically, we employ schedulers with exponential decay rate of the form $\eta(n) = \eta_0 \times e^{-d\,n}$, where $\eta_0$ is a fixed initial value

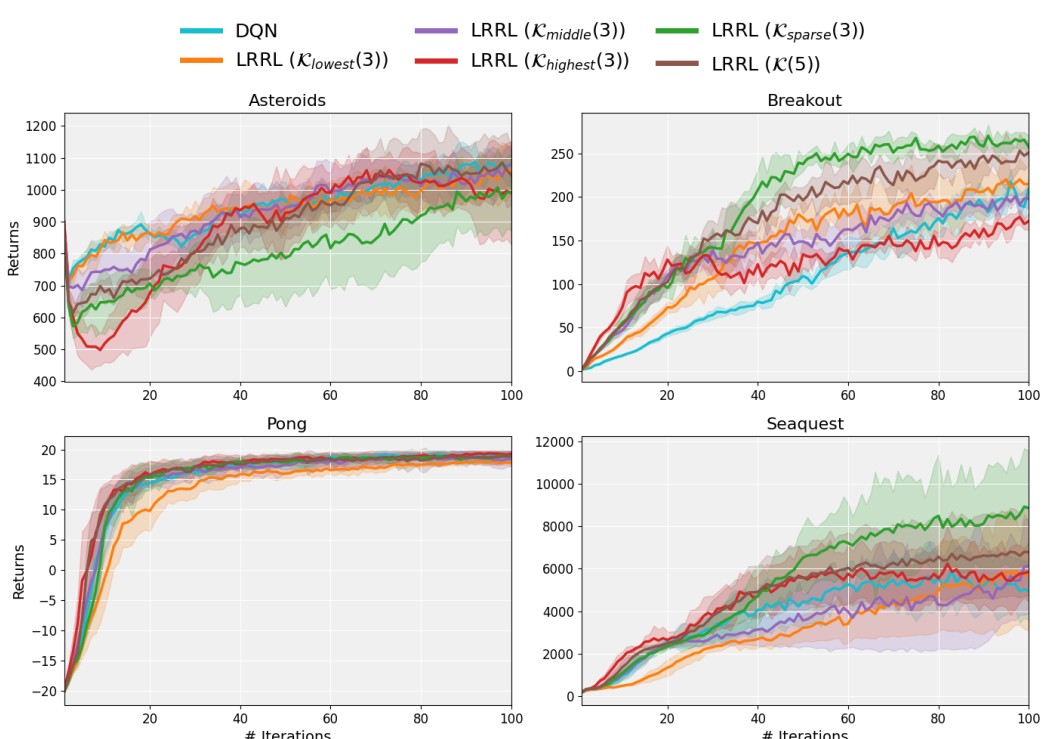

Figure 1: A comparison between 5 configurations of LRRL with various learning rate subsets and the DQN algorithm reaching the best performance among possible learning rates.

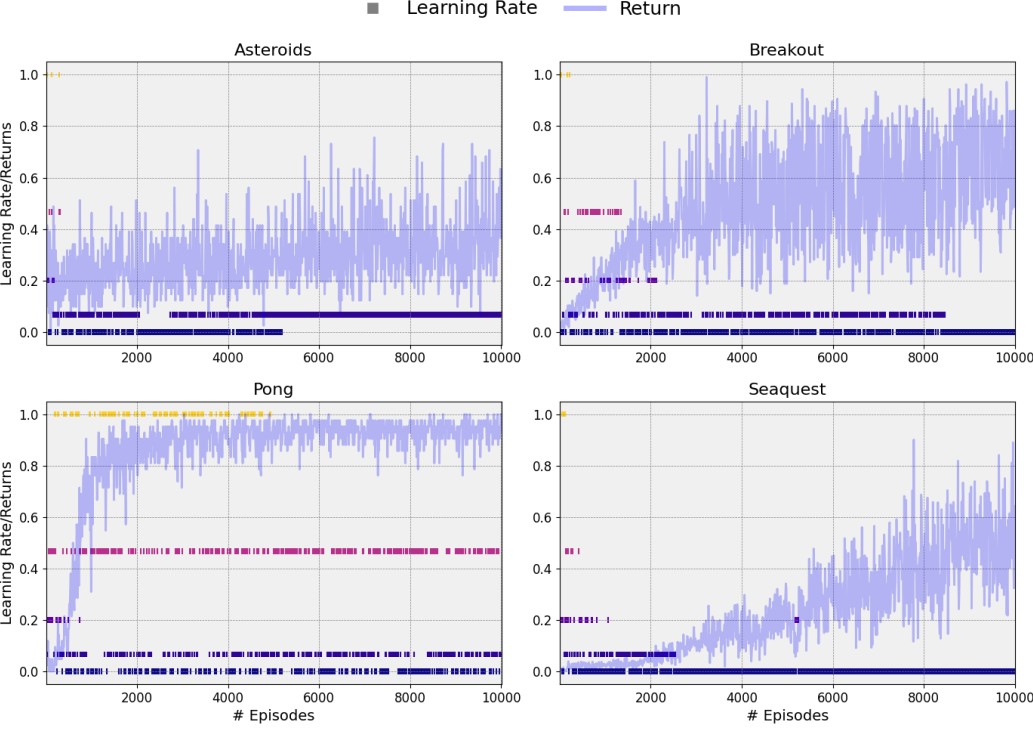

Figure 2: Systematic sampling of normalized learning rates and returns over the training steps using LRRL $\mathcal{K}(5)$ with Adam optimizer, through a single run. For each episode, we show the selected learning rate using different colors.

(common to each scheduler and equal to $6.25 \times 10^{-5}$ in our experiment), $d$ is the exponential decay rate and $n$ is the number of policy updates (*i.e.*, of MAB rounds). We define a set of 3 schedulers $\mathcal{K}_s$, where each arm represents a scheduler using a different decay rate $d = \{1, 2, 3\} \times 10^{-7}$, and compare the results of LRRL with each scheduler individually, using the Adam optimizer.

Figure 3 and Table 2 show that LRRL combined with schedulers can substantially increase final performance compared to using exponential decay schedulers for some environments while remaining competitive for others. The dashed black line represents the max average return achieved by Adam without learning rate decay, resulting in slightly worse performance compared to using schedulers, aligning with findings in previous work by Andrychowicz et al. (2021), who linearly decay the learning rate to 0.

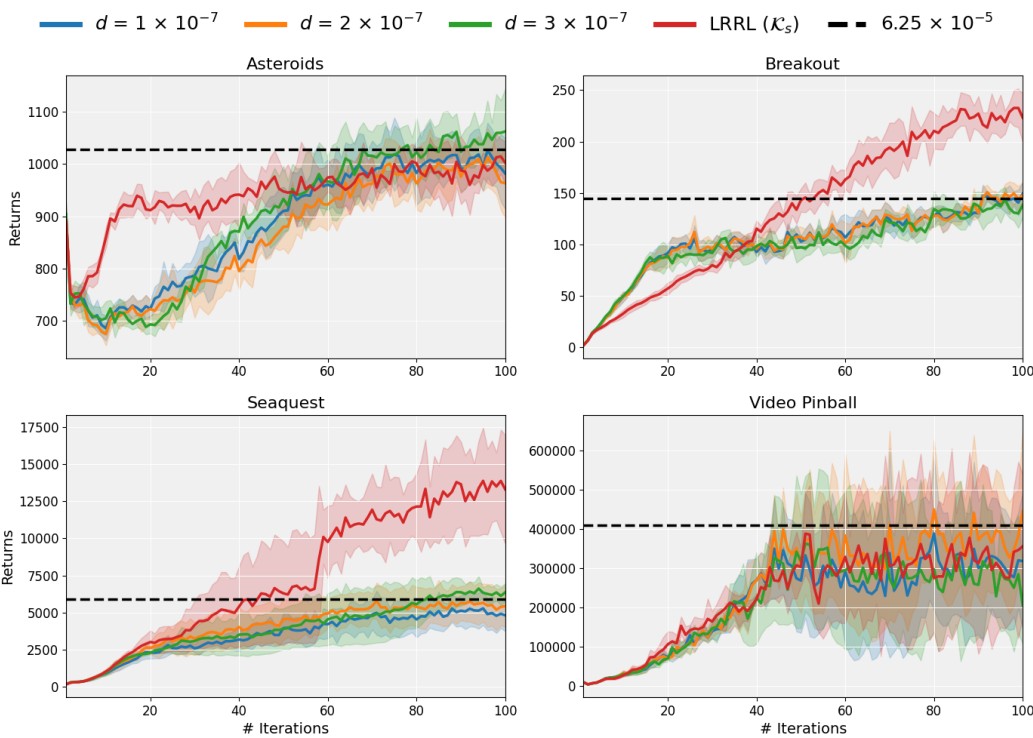

Figure 3: A comparison between LRRL with arms as schedulers and schedulers individually.

| Game | DQN | $d = 1 \times 10^{-7}$ | $d = 2 \times 10^{-7}$ | $d = 3 \times 10^{-7}$ | **LRRL** ($\mathcal{K}_s$) |
|---|---|---|---|---|---|
| Asteroids | $1\,028 \pm 53$ | $1\,025 \pm 52$ | $1\,013 \pm 56$ | $\mathbf{1\,063} \pm 82$ | $1\,015 \pm 33$ |
| Breakout | $144 \pm 12$ | $149 \pm 11$ | $151 \pm 7$ | $144 \pm 12$ | $\mathbf{233} \pm 19$ |
| Seaquest | $5\,881 \pm 1\,533$ | $5\,284 \pm 1\,134$ | $5\,793 \pm 1\,225$ | $6\,612 \pm 835$ | $\mathbf{13\,864} \pm 3\,581$ |
| Video Pinball | $410\,186 \pm 193\,328$ | $388\,768 \pm 195\,150$ | $\mathbf{450\,015} \pm 178\,876$ | $362\,645 \pm 172\,169$ | $388\,308 \pm 103\,862$ |

Table 2: Max average return (best in **bold**) and its standard deviation for 4 Atari games.

## 5.3 RMSPROP OPTIMIZER AND MORE ENVIRONMENTS

Another widely used optimizer for training deep RL models is RMSProp, which, like Adam, features an adaptive learning rate mechanism. Adam builds upon RMSProp by retaining exponential moving averages to give more weight to recent gradients while incorporating momentum. Although the standard RMSProp does not feature momentum, we found that adding momentum to RMSProp can increase both the performance of DQN and LRRL, aligning with findings in the literature (Qian, 1999; Andrychowicz et al., 2021).

In the following experiment, we compare the performance of RMSProp with Nesterov's momentum (RMSProp-M), and Adam when coupled with either LRRL or the best-performing single learning

rate when using DQN. As shown in Figure 4 and Table 3, LRRL coupled with Adam consistently outperforms our configuration using RMSProp-M and the baseline using standard DQN. Moreover, LRRL (RMSProp-M) underperforms compared to DQN without LRRL in two out of three tasks due to its slow convergence despite better jumpstart performance. Future work should investigate whether this slow convergence is linked to factors such as the environment's stochasticity or the optimizer's features such as the absence of bias correction in the first and second moment estimates.

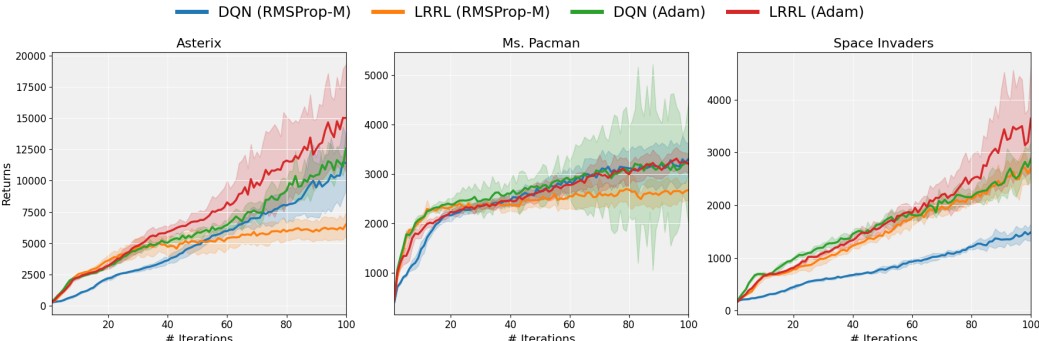

Figure 4: A comparison between Adam and RMSProp with momentum, using either DQN or LRRL.

| Game | DQN (RMSProp-M) | LRRL (RMSProp-M) | DQN (Adam) | LRRL (Adam) |
|---|---|---|---|---|
| Asterix | $11\,464 \pm 2\,848$ | $6\,499 \pm 927$ | $12\,561 \pm 1\,245$ | $\mathbf{15\,017} \pm 3\,892$ |
| Ms. Pacman | $3\,301 \pm 310$ | $2\,696 \pm 248$ | $3\,232 \pm 114$ | $\mathbf{3\,310} \pm 231$ |
| Space Invaders | $1\,490 \pm 132$ | $2\,712 \pm 73$ | $2\,874 \pm 319$ | $\mathbf{3\,641} \pm 1\,030$ |

Table 3: Max average return (best in **bold**) and its standard deviation for 3 Atari games.

## 6 CONCLUSION

In this work, we introduced dynamic Learning Rate for Deep Reinforcement Learning (LRRL), a meta-learning approach for selecting the optimizer's learning rate on the fly. We demonstrated empirically that combining LRRL with the Adam optimizer could significantly enhance the performance of the value-based algorithm DQN, outperforming baselines and learning rate schedulers in some tasks while remaining competitive in others. Furthermore, by employing a multi-armed bandit algorithm, LRRL reduces the need for extensive hyperparameter tuning, as it explores a set of learning rates in a single run with minimal extra computational overhead.

While this work focused on dynamically selecting the best-performing learning rate, future investigations could extend LRRL ideas to other critical hyperparameters, such as mini-batch size, which also plays a key role in the model's convergence. Moreover, although LRRL selects learning rates based on policy performance, alternative feedback mechanisms could be explored, such as using gradient information to select block-wise (*e.g.*, per-layer) learning rates, extending these ideas to supervised learning and applying them to other non-stationary objective functions, including those encountered in Continual Learning (Rusu et al., 2016; Kirkpatrick et al., 2016; Abel et al., 2023).

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

## A  SUPPLEMENTARY EXPERIMENTS

### A.1  BASELINE EVALUATION WITH VARYING LEARNING RATES

To establish a baseline to compare learning without our approach LRRL, we run individual arm values as baseline learning rate using the Adam optimizer. The results presented in Figure 5 align with common expectations by showing that higher learning rates fail to learn for most environments while lower ones can lead to the worst jumpstart performance and slow convergence.

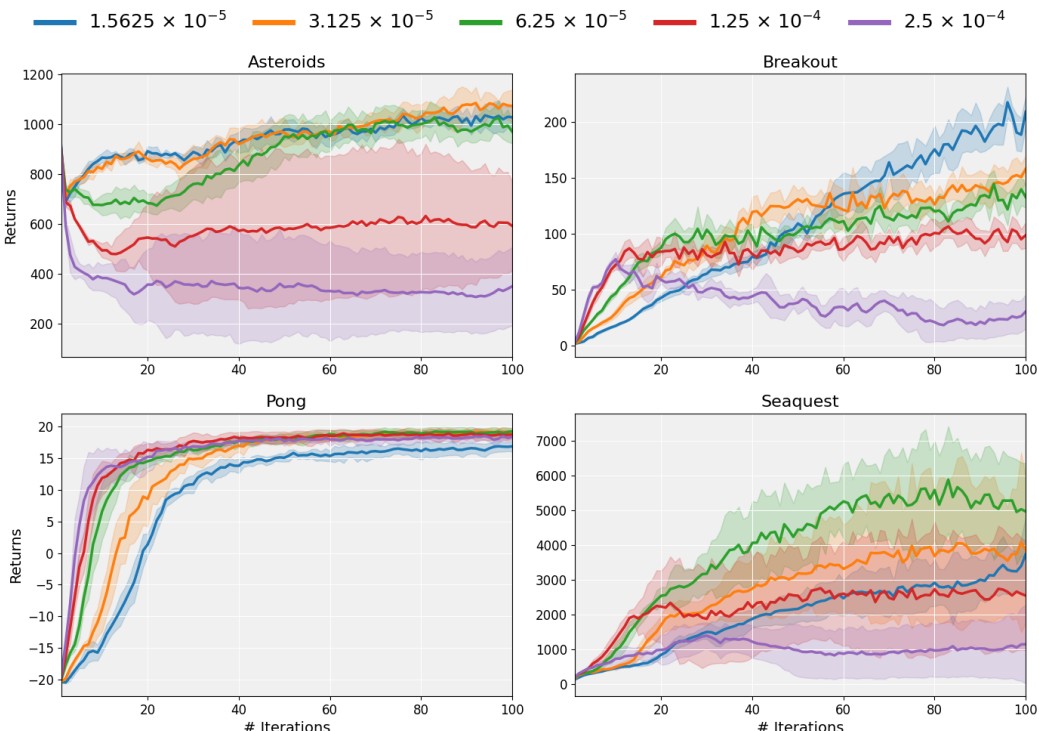

Figure 5: DQN performance using Adam optimizer with varying learning rates across 4 Atari games.

### A.2  A COMPARISON BETWEEN MULTI-ARMED BANDITS ALGORITHMS

Adversarial MAB algorithms are designed for environments where the reward distribution changes over time. In contrast, stochastic MAB algorithms assume that rewards are drawn from fixed but unknown probability distributions. To validate our choice and address our method's robustness, we compare Exp3 with the stochastic MAB algorithm MOSS (Minimax Optimal Strategy in the Stochastic case) (Audibert & Bubeck, 2009). MOSS trade-off exploration-exploitation by pulling the arm $k$ with highest upper confidence bound given by:

$$B_k(n) = \hat{\mu}_k(n) + \rho \sqrt{\frac{\max\left(\log \frac{n}{K n_k(n)}, 0\right)}{n_k(n)}}$$

where $\hat{\mu}_k(n)$ is the empirical average reward for arm $k$ and $n_k(n)$ is the number of times it has been pulled up to timestep $t$.

In Figure 6, we use different bandit step-sizes $\alpha$ for Exp3 and parameter $\rho$, which balance exploration and exploitation in MOSS. Additionally, the bandit feedback used in MOSS is the average cumulative reward $\frac{R}{C}$. The results indicate that while Exp3 performs better overall, MOSS can still achieve competitive results depending on the amount of exploration, demonstrating the robustness of our approach regarding the MAB algorithm employed.

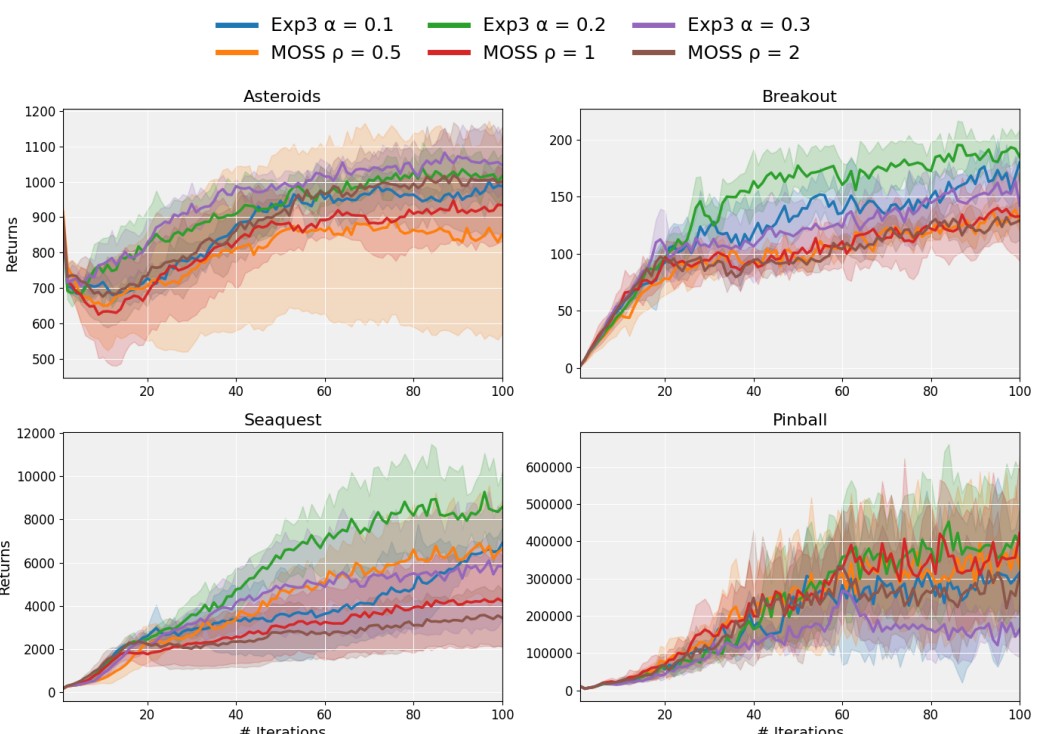

Figure 6: A comparison between adversarial and stochastic MAB algorithms.

# B    EXPERIMENTS DETAILS

## B.1    EVALUATION TERMINOLOGY

In this section, we describe the evaluation metrics that can be used to evaluate agent's performance as it interacts with an environment. Based on Dopamine, we use the evaluation step-size "iterations", which is defined as a predetermined number of episodes. Figure 7 illustrates the evaluation metrics used in this work, as defined in (Taylor & Stone, 2009):

- **Max average return:** The highest average return obtained by an algorithm throughout the learning process. It is calculated by averaging the outcomes across multiple individual runs.

- **Final performance:** The performance of an algorithm after a predefined number of inter-actions. While two algorithms may reach the same final performance, they might require different amounts of data to do so. This metric captures the efficiency of an algorithm in reaching a certain level of performance within a limited number of interactions. In Fig-ure 7, the final performance overlaps with the max average return, represented by the black dashed line.

- **Jumpstart performance:** The performance at the initial stages of training, starting from a policy with randomized parameters $\theta$. In Figure 7, Algorithm B exhibits better jumpstart performance but ultimately achieves lower final performance than Algorithm A. A lower jumpstart performance can result from factors such as a lower learning rate, although this work demonstrates that this does not necessarily lead to worse final performance.

## B.2    FURTHER EXPERIMENTAL DETAILS

In the following, we list the set of arms, optimizer and the bandit step-size used in each experiment.

**Section 5.1 – Comparing LRRL with Standard Learning.**

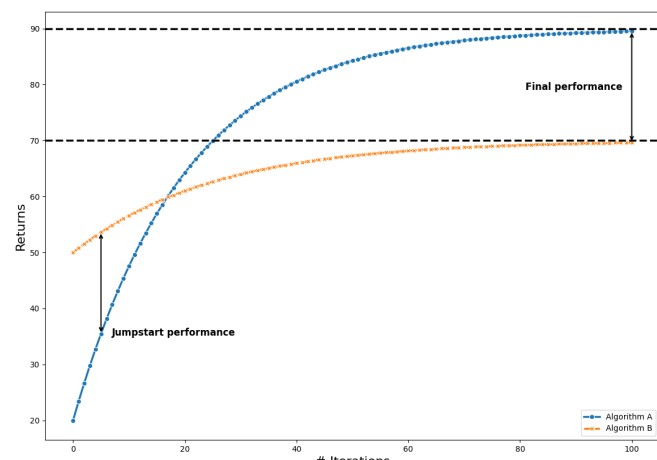

Figure 7: Performance curves of two RL algorithms (adapted from Taylor & Stone (2009)).

- Optimizer: Adam
- Bandit step-size: $\alpha = 0.2$
- Considered sets of learning rates:

$$\mathcal{K}(5) = \left\{ 1.5625 \times 10^{-5}, 3.125 \times 10^{-5}, 6.25 \times 10^{-5}, 1.25 \times 10^{-4}, 2.5 \times 10^{-4} \right\}$$

$$\mathcal{K}_{\text{lowest}}(3) = \left\{ 1.5625 \times 10^{-5}, 3.125 \times 10^{-5}, 6.25 \times 10^{-5} \right\}$$

$$\mathcal{K}_{\text{middle}}(3) = \left\{ 3.125 \times 10^{-5}, 6.25 \times 10^{-5}, 1.25 \times 10^{-4} \right\}$$

$$\mathcal{K}_{\text{highest}}(3) = \left\{ 6.25 \times 10^{-5}, 1.25 \times 10^{-4}, 2.5 \times 10^{-4} \right\}$$

$$\mathcal{K}_{\text{sparse}}(3) = \left\{ 1.5625 \times 10^{-5}, \qquad\quad 6.25 \times 10^{-5}, \qquad\quad 2.5 \times 10^{-4} \right\}$$

**Section 5.2 – Combining and Comparing Schedulers with LRRL.**

- Optimizer: Adam
- Bandit step-size: $\alpha = 0.2$
- Initial learning rate of schedulers: $\eta_0 = 6.25 \times 10^{-5}$

**Section 5.3 – RMSProp Optimizer and More Environments.**

- Optimizer: RMSProp for baselines
- Bandit step-size: $\alpha = 0.2$
- LRRL set of learning rates: $\mathcal{K} = \left\{ 1.5625 \times 10^{-5}, 3.125 \times 10^{-5}, 6.25 \times 10^{-5} \right\}$

**Section A.2 – A Comparison between Multi-Armed Bandit Algorithms.**

- Optimizer: Adam
- MOSS/Exp3 set of learning rates: $\mathcal{K} = \left\{ 3.125 \times 10^{-5}, 6.25 \times 10^{-5}, 1.25 \times 10^{-4} \right\}$

### B.3 HYPERPARAMETERS

In this section, we outline the hyperparameters used in the experiments. The optimizers from the Optax library (DeepMind et al., 2020) are employed alongside the JAX (Bradbury et al., 2018) implementation of the DQN algorithm (Mnih et al., 2013; 2015), as provided by the Dopamine framework (Castro et al., 2018).

| Hyperparameter | Setting |
|---|---|
| Sticky actions | True |
| Sticky actions probability | 0.25 |
| Discount factor ($\gamma$) | 0.99 |
| Frames stacked | 4 |
| Mini-batch size ($\mathcal{B}$) | 32 |
| Replay memory start size | 20000 |
| Learning network update rate ($\lambda$) | 4 steps |
| Minimum environment steps ($\kappa$) | 1 episode |
| Target network update rate ($\tau$) | 8000 steps |
| Initial exploration ($\epsilon$) | 1 |
| Exploration decay rate | 0.01 |
| Exploration decay period | 250000 steps |
| Environment steps per iteration | 250000 steps |
| Reward clipping | [-1, 1] |
| Network neurons per layer | 32, 64, 64 |
| Hardware | V100 GPU |
| **Adam hyperparameters** | |
| $\beta_1$ decay | 0.9 |
| $\beta_2$ decay | 0.999 |
| Eps | 1.5e-4 |
| **RMSProp hyperparameters** | |
| Decay | 0.9 |
| Momentum (if *True*) | 0.999 |
| Centered | False |
| Eps | 1.5e-4 |

Table 4: Hyperparameters

