# OpenReview forum: "Dynamic Learning Rate for Deep Reinforcement Learning: A Bandit Approach"
_ICLR.cc/2025/Conference — Submitted to ICLR 2025_

### Official Review · Reviewer_ZFf4 · 2024-11-03

**Soundness:** 2
**Presentation:** 3
**Contribution:** 1
**Rating:** 3
**Confidence:** 5

**Summary:**

This work presents Dynamic Learning Rate for Reinforcement Learning (LRRL), a meta-learning approach that uses a multi-armed bandit algorithm to dynamically adjust the learning rate of an RL agent during training. Each arm of the bandit represents a different learning rate, and the bandit's feedback mechanism uses the cumulative returns of the RL policy to update the probability distribution over the arms. The authors argue that this approach addresses the non-stationarity of the objective function in RL, where the optimal learning rate can change throughout training, and claim that it reduces the need for extensive hyperparameter tuning compared to traditional learning rate decay schedules. Empirical results on several Atari games using the DQN algorithm are presented, comparing LRRL against fixed baselines, traditional schedulers, and alternative optimizers.

**Strengths:**

Overall, adjusting the learning rate during training is an important problem. The proposed approach is conceptually straightforward, using multi-armed bandits, which is also relatively easy to implement.

**Weaknesses:**

The main idea in the paper for using bandits for hyperparameter selection in RL is not new. Multiple prior works have explored similar ideas, using bandits to adapt various aspects of learning, including exploration strategies and policy updates. The authors' attempt to position LRRL as a distinct approach within the context of meta-learning is also not entirely convincing, as meta-gradient methods already address the adaptation of learning parameters (see e.g., Flennerhag, Sebastian, et al., among many others).

Re experiments: The scope of the experiments in the paper is limited. The authors focus on a single RL algorithm (DQN) and a restricted set of environments. I propose the authors run their approach using sota RL algorithms, as well as continuous control tasks, which would help understand the broader applicability of LRRL. Finally, the authors should include a comparison of their approach to sota meta-gradient method.

The paper could also benefit from a more detailed discussion of the theoretical properties of Exp3 in the context of learning rate adaptation and a more comprehensive exploration of alternative bandit algorithms (e.g., linear contextual bandits).

**Questions:**

See above.

---

> ### Author Response · Authors · 2024-11-22
>
> Thank you for your review and feedback on the distinctiveness of LRRL and the scope of our experiments. We address your comments below:
>
> - **LRRL as a distinct approach**: We believe LRRL is distinct within meta-learning by employing multi-armed bandits. While prior work employs Successive Halving and infinite arms for hyperparameter searches, LRRL aligns more closely with scheduling approaches by using an adversarial algorithm with RL policy's performance as feedback to select the learning rate on the fly.
>
> - **Single RL algorithm and limited environments**: We acknowledge that testing with additional algorithms would strengthen our claims. However, many algorithms build on the core principles of DQN, which supports the generality of our findings. Regarding environments, we selected a subset of games to test due to their computational cost, considering various challenges, such as reward sparsity and exploration requirements, by looking at their learning curve.

---

### Official Review · Reviewer_sN7H · 2024-11-04

**Soundness:** 2
**Presentation:** 4
**Contribution:** 2
**Rating:** 3
**Confidence:** 3

**Summary:**

The paper presents a bandit algorithm for selecting the learning rate of DQN in a set of Atari games. The paper shows how the bandit-selected learning rates over one fixed set of learning rates can, in some cases, yield better performance than the default DQN learning rate.

**Strengths:**

* Clear Writing, easy to follow and understand
* Good empirical description. Almost all the necessary details about how the experiment was run were included in the main body or Appendix B

**Weaknesses:**

* The biggest weakness is the range of learning rates tested. The paper uses a set of 5 learning rates, all within one order of magnitude, to test LRRL. Most deep RL algorithms are highly sensitive to learning rates. While these are known to be a good range for DQN, as it has been around for over a decade, it is difficult to say whether LRRL equipped with this range will work well with other new algorithms we want to deploy this one. Additionally, would the Adam vs RMSProp results still hold if different ranges of learning rates were given? Two suggestions could address this:
1. Different sets of learning rates for LRRLs.
     - A wider range; Like something that goes from 10^-1 to 10^-6
     - Randomly generated ranges. Maybe have uniform or log uniform sampling over some range 10^-1 to 10^-6, and see how LRRL works with 5 LRs in this range
2. Try with other algorithms (see next bullet point)

* The other big concern with this paper is that it makes claims for Deep Reinforcement Learning Algorithms but only provides evidence on one (DQN). Two fixes could be either:
     - Try LRRL with learning rates on other algorithms: PPO, SAC, A3C, TD3, DDQN etc.
     - Re-adjust the claims in the paper to make it clear that you are studying LRRL+DQN. This would include changes in wording throughout and changing the title to Dynamic Learning Rate for DQN.

* Since the experiments section uses 5 independent runs to measure the average return, we would recommend a different uncertainty measure for the reported results:
     - A bootstrapped confidence interval or a tolerance interval that could be representative of the variability in performance across different seeds and the same learning rate. This would be useful in figure 1.
    - 95% confidence intervals. Although this comes with an assumption that the returns for your given agent+learning rate is normally distributed.
    - The meaning shaded region was not explicitly mentioned any where. I assume it is std dev?

* It would have been useful to see how a bandit algorithm over this set K(5) would compare to other hyperparameter approaches such as bayesian optimization packages like Optuna, which is  commonly used for dynamically tuning learning rates and other deep reinforcement learning hyperparameters.

**Questions:**

Why was Pong swapped out for Video PinBall in section 5.2? How did the paper choose which Atari games to use for each section? Did the paper look at all Atari environments but reported those with a significant result in the main body? If so, this should be made clear in the main body.

---

> ### Author Response · Authors · 2024-11-22
>
> Thank you for your review and detailed questions regarding learning rate sets, algorithms, and environments. We address your concerns below:
>
> - **Learning rate sets in LRRL**: The chosen learning rate set is centered on default values in Dopamine to enable comparisons with other algorithms. However, as shown in Figure 5, some learning rates considered in the arms set fail to perform well specific tasks, which give us insights about the LRRL robustness. Nevertheless, the reviewer suggestion of exploring LRRL performance with logarithmic sampling sets, which spans several orders of magnitude, would be an interesting direction.
>
> - **Evaluating alternative algorithms**: We chose DQN as our baseline because many other algorithms are built upon it, minimizing the confounding factors that more advanced methods might introduce during assessment. Nonetheless, we acknowledge that testing LRRL with additional algorithms could further strengthen the reliability of our findings.
>
> - **Optuna comparison**: Optuna [1] appears well-suited for hyperparameter optimization, incorporating methods like Successive Halving. We referenced similar approaches in our Related Works section, highlighting their distinction regarding LRRL.
>
> - **Atari environments**: We define a subset of Atari games to test due to their computational cost, selecting them by looking into their learning curves from Dopamine baselines. These curves provided insights into reward sparsity, stochasticity, and exploration demands. While Pong highlights a particular case of pulled arms/returns, we replace it with a different environment since most of the chosen learning rates could perform the task relatively well.
>
> *[1] Takuya Akiba et al. 2019. Optuna: A Next-generation Hyperparameter Optimization Framework. In Proceedings of the 25th ACM SIGKDD International Conference on Knowledge Discovery & Data Mining (KDD '19), pp. 2623–2631. https://doi.org/10.1145/3292500.3330701*

---

### Official Review · Reviewer_DBUv · 2024-11-04

**Soundness:** 3
**Presentation:** 3
**Contribution:** 2
**Rating:** 3
**Confidence:** 3

**Summary:**

The paper proposes to use a non-stationary version of Exp3 (a multi-armed bandit algorithm) to dynamically adjust the learning rate in deep reinforcement learning models during training. The algorithm begins by initializing a set of learning rates. During training, the Exp3 algorithm selects a learning rate from the set based on the agent’s recent performance. The agent then interacts with the environment for a fixed number of steps using the selected learning rate, before the process repeats.

**Strengths:**

* The authors show that LRRL achieves competitive performance compared to standard deep RL algorithms that use fixed learning rates or traditional learning rate schedulers. They test the algorithm using the DQN algorithm with the Adam optimizer on a variety of Atari games.
* LRRL reduces the need for hyperparameter optimization in principle, but see below for my concern about this.

**Weaknesses:**

* The choice of the number of learning rate arms and their values still requires task-specific tuning. This is evidenced in Table 1, where LRRL seems to have significantly different performance depending on what choice of learning rates are used. I would have expected the bandit algorithm to be relatively robust to its set of arms, but this is not the case.
    * K_sparse(3) performs far better than the other choices. In my view, this approach seems to just shift hyperparameter tuning to a different parameter.

* One intuition for decay of learning rates is that the optimizer takes large steps toward the optimal policy first (perhaps incurring less reward), but later on "hones in" on the best policy using a smaller learning rate (incurring high reward). The bandit approach wouldn't be able to encode this kind of intuition, where you sacrifice some reward initially so that higher rewards can be attained later on. I think this is a main drawback of this approach.

**Questions:**

* How would the authors address the concern of having to tune the set of learning rates for Exp3?

* Would Exp3 be able to encode the intuition I described above in some way? How would the authors reconcile this potential drawback?

---

> ### Author Response · Authors · 2024-11-22
>
> Thank you for your review and insightful questions about the potential drawbacks of LRRL. Below, we provide our responses:
>
> - **How would the authors address the concern of having to tune the set of learning rates for Exp3?** Although defining the arms set remains necessary, LRRL mitigates the exhaustive search required by traditional grid search by continuously adapting the learning dynamically. As shown in Figure 5, some of the learning rates in the set cannot perform some tasks well, while Figure 1 highlights the importance of defining an appropriate set size and its values. Since LRRL is a multi-armed bandit-based framework, it inherits the exploration-exploitation trade-off, enabling it to balance exploring less-tested rates and exploiting those that perform well along with the training steps.
>
> - **Would Exp3 be able to encode the intuition I described above...?** LRRL balances exploration and exploitation, as evidenced in Figure 2, where it mirrors scheduler behavior by sampling higher learning rates early on and later only smaller ones. Incorporating contextual bandits, as suggested by other reviewers, could potentially enhance this framework by leveraging context features such as the cumulative number of steps/epochs.

---

> > ### Comment · Reviewer_DBUv · 2024-11-25
> > **Thank you for your comment**
> >
> > Hi, thank you for your response. I will keep my score.

---

### Official Review · Reviewer_kWKK · 2024-11-04

**Soundness:** 3
**Presentation:** 3
**Contribution:** 2
**Rating:** 5
**Confidence:** 4

**Summary:**

This paper proposes a meta-learning algorithm, LRRL, which selects the learning rate throughout the training process using a bandit algorithm. This algorithm is tested in with DQN on deep RL benchmark environments and in conjunction with differnet optimizers such as RMSProp and Adam. Some analysis is done to assess the learning rate choices through learning.

**Strengths:**

The paper presents the proposed algorithm quickly and the motivation is clear.
I find that the formulation of the meta-learning problem as a bandit problem is interesting and it is nice to see this direction explored more in this paper.

There are some interesting insights into which learning rates are favourable for different environments by looking at the ones chosen by the proposed algorithm (in Fig.2).

**Weaknesses:**

Generally, my main concern are the purported benefits of the algorithm.
The original motivation is that learning rates can be important to tune in general and also throughout training.
From the results (e.g. Fig.1), it looks like the choice of the set of learning rates for LRRL is important.
Many of the settings do not lead to any benefit over the baseline agent. Moreover, simply choosing the largest set of learning rates, $\mathcal{K}(5)$, does not necessarily lead to any improvements.
This runs counter to the original motivation of reducing hyperparameter sensitivity to the learning rate.

**Questions:**

- Tables 1 and 2 which report max average returns seem unecessary given that the entire learning curves are also presented.
Also, reporting the maximum return is generally not a good practice due to the additional noise in the estimate. See [1] for a more in-depth discussion of this.

- In section 5.1, which optimizer is used with DQN? Is it RMSProp, Adam or a different choice?

- In fig.1, what are "# iterations" on the x-axis? How many training steps are done in total?

- When formulating the meta-learning problem as a bandit problem, the formulation does not make use of the fact that different learning rates are related. In this paper, the learning rates are treated as separate discrete actions while they are in fact continuous quantities. Similar learning rates should have similar optimization properties.
Perhaps using a formulation that allows some generalization between the bandit actions (learning rates) could be useful here. i.e. linear bandits or more general contextual bandits.


- Because of the way the reward is structured, it seems like sparse rewards would be a problem for these methods.
If a feedback window happens to be in the middle of the episode where no rewards are given, then it would make it seem like that meta-action was not useful.
That could explain why in Pong, which has a relatively sparse reward, the agent does not manage to adapt the learning rate much and ends up mostly picking learning rates relatively uniformly.

- As a comment: In the background section (lines 177 onward), it is mentioned that the adversarial bandit framework deals with nonsationarity, but this is not quite accurate. For that, you would need to consider sequences of changing best actions, not just one action in hindsight. For a more detailed discussion, see Ch 31.1 from [2]. I would suggest rewording this section a little.
This does not negatively impact the algorithm in the end since the actual bandit algorithm used does have a decay parameter which does try to account for the nonstationarity by emphasizing recent experiences.


[1] "Deep Reinforcement Learning that Matters" Henderson et al.

[2] "Bandit Algorithms" Lattimore and Szepesvari (Chapter 31.1)  https://tor-lattimore.com/downloads/book/book.pdf

---

> ### Author Response · Authors · 2024-11-22
>
> Thank you for your thoughtful review and questions regarding the proposed method. Below, we address your concerns in the order they were raised:
>
> - **Figures and Tables**: We agree that presenting both figures and tables can be redundant. In the revised version of the paper, we will remove the tables to enhance clarity.
>
> - **The optimizer in Section 5.1**: Adam is employed as the optimizer. While this information is available in Appendix B, we will mention it in the main body for completeness.
>
> - **Number of iterations**: In Dopamine, the number of iterations is defined as 1 million frames per iteration. We acknowledge that this important detail was overlooked, and we will ensure it is clearly stated in the revised version.
>
> - **Arms are related**: We acknowledge that arms can exhibit correlations, and employing a contextual MAB framework could explicitly model these relationships, potentially extending LRRL. However, although in a distinct context, we believe that our current feedback mechanism implicitly encodes a temporal dependency. Specifically, computing the bandit feedback as the difference between the current arm's performance and the average feedback of the past $n$ arms pulled. This mechanism ensures that the evaluation of each arm reflects its relative effectiveness compared to recent choices, thereby capturing a form of correlation within the arm set.
>
> - **Sparse rewards** are indeed challenging for LRRL, as it relies on the RL policy outcomes to update arms probabilities. However, note that the variable $\kappa$ in Algorithm 1 addresses this by controlling the number of steps before updating the learning rate. This allows for multiple updates using the same learning rate before adjusting the arms probabilities.
>
> - **Regret definition**: Thanks for pointing out the misconception between our regret definition and the non-stationary setting. As pointed out by the reviewer, the Exp3 version used for LRRL does account for non-stationary, although its theoretical guarantees regarding the regret in that setting are still to be established (to the best of our knowledge). We will clarify it on the revised version.

---

### Meta-Review · Area_Chair_cZ93 · 2024-12-16

**Metareview:**

This work proposes a meta learning approach to learning rates selection for reinforcement learning algorithms based on the Multi Armed Bandits model. Although it was agreed that this approach holds promise for the problem of hyperparameter selection, the reviewers also agreed that the experimental evaluation of this work is insufficient to show substantiated promise of the proposed approach. We encourage the authors to expand this evaluation to larger and more convincing scenarios. Additionally, this work would benefit from drawing connections with other bandit literature works such as online model selection techniques that can also be used to solve the problem tackled by the authors.

**Additional Comments On Reviewer Discussion:**

The reviewers agreed that the contributions of this work as presented do not present substantial benefits in the learning rate selection problem. The experimental results did not substantiate the authors claims. This does not mean the methods do not show promise, but that the execution in the current manuscript is not yet showing the promised results.

---

### Decision · Program_Chairs · 2025-01-22

Reject